# Josephson emission with frequency span 1–11 THz from small $Bi_2Sr_2CaCu_2O_{8+\delta}$ mesa structures

E.A. Borodianskyi[1] & V.M. Krasnov [1]

Mesa structures made of $Bi_2Sr_2CaCu_2O_{8+\delta}$ high-temperature superconductor represent stacks of atomic scale intrinsic Josephson junctions. They can be used for generation of high-frequency electromagnetic waves. Here we analyze Josephson emission from small-but-high mesas (with a small area, but containing many stacked junctions). We have found strong evidence for tunable terahertz emission with a good efficacy in a record high-frequency span 1–11 THz, approaching the theoretical upper limit for this superconductor. Emission maxima correspond to in-phase cavity modes in the mesas, indicating coherent superradiant nature of the emission. We conclude that terahertz emission requires a threshold number of junctions $N \sim 100$. The threshold behavior is not present in the classical description of stacked Josephson junctions and suggests importance of laser-like cascade amplification of the photon number in the cavity.

[1] Department of Physics, Stockholm University, AlbaNova University Center, SE—106 91 Stockholm, Sweden. Correspondence and requests for materials should be addressed to V.M.K. (email: Vladimir.Krasnov@fysik.su.se)

Progress in terahertz (THz) science and technology requires proper THz sources in the 0.1–10 THz range[1]. Pulsed broad-band THz sources are fairly well developed today[2–4]. However, it is much more difficult to built compact, continuous-wave, narrow line-width THz lasers, which would also be tunable in a broad frequency range. Significant progress in this direction is achieved by semiconducting quantum cascade lasers[5,6]. A limited tunability of such lasers can be partially obviated by multi-mode frequency comb operation, allowing a factor-two (one octave) frequency span[7], or by post-processing[8]. Using superconductors instead of semiconductors provide an alternative technology for creation of inherently highly tunable continuous-wave, narrow line-width, compact THz sources.

Superconducting Josephson junctions allow a direct conversion of DC-voltage into high-frequency electromagnetic oscillations with a tunable frequency[9–13]

$$f = \frac{2e}{h} V \quad (\simeq 0.48359 \, \text{THz/mV}). \tag{1}$$

Here $V$ is the voltage per junction, $e$—electron charge and $h$—Plancks constant. The frequency is limited only by the superconducting energy gap, which is sub-THz for low-$T_c$[10] and $\geq$10 THz for high-$T_c$ superconductors[14]. Of particular interest are layered $Bi_2Sr_2CaCu_2O_{8+\delta}$ (Bi-2212) cuprates, which represent natural stacks of atomic scale intrinsic Josephson junctions (IJJs)[15]. Bi-2212 mesa structures allow integration of a large amount of junctions, required for coherent superradiant enhancement of the emission power[9]. A significant sub-THz emission has been reported from large Bi-2212 mesas[11,12,16–18]. But further enhancement of the emission power and frequency is hindered by Joule heating[17,18], which limits the bias voltage and the primary (first harmonic) oscillation frequency[18–20].

It has long been anticipated that small Bi-2212 mesas may have many benefits as THz oscillators: (i) Edge effects[21,22] and capacitive coupling[23] persuade in-phase synchronization of junctions, needed for superradiance; (ii) Self-heating is reduced proportional to the mesa size[24], which allows operation at high voltages and frequencies; (iii) The frequency and the quality factor $Q$ of the primary geometrical resonance (cavity mode) increase inversely proportional to the mesa size. Fiske steps with $Q > 100$ were reported for µm-size Bi-2212 mesas[25]. This should strongly boost the emission efficiency because high-$Q$ resonances enhance the emission power $\propto Q^2$ and decrease the line-width $\propto 1/Q$[26,27]. (iv) Small mesas can be made free from defects and with identical junctions, simplifying their synchronization. However, experimental realization of small area Bi-2212 oscillators has been proven very difficult. After initial encouraging reports[28,29], several groups reported negative results. For example, sub pW level power was reported in ref.[30] from small mesas, which contained less than 60 junctions. This is puzzling because theoretical analysis univocally predicted significant emission from small mesas[23,26,27,31,32].

Here we study Josephson emission from small-but-high Bi-2212 mesas, with µm in-plane size, but with a large number of stacked junctions $N > 100$. We have found strong evidence for tunable electromagnetic wave emission in a record high-frequency span 1–11 THz at the primary (first harmonics) Josephson frequency. The highest frequency is close to the theoretical upper limit for this high-$T_c$ superconductor. From analysis of size-dependence of the emission spectra we conclude that the emission maxima correspond to in-phase cavity modes in the mesas, pointing towards coherent superradiant nature of the observed emission. We argue that the superradiant emission from Bi-2212 mesas requires a threshold number of synchronized junctions $N > 100$. This may indicate presence of a novel laser-like mechanism of emission.

## Results

**Sample characterization**. Figure 1a, b show 1a an image and 1b a three-dimensional sketch of the studied sample. It consists of nine mesas with attached metallic electrodes. Details of sample fabrication and the experimental setup can be found in the Supplementary Notes 1 and 2. Each of the mesas can be biased independently and can be used as either a generator or a detector[29,33,34]. Below we will show results for generator/detector configurations 2/5 and 4/1 marked in Fig. 1a. The two generators have considerably different shapes. The mesa #2 is almost square shaped ($5.2 \times 4.6 \, \mu m^2$) and the mesa #4 is elongated ($11.5 \times 2.6 \, \mu m^2$). In both cases we use the smallest distant mesas as detectors mesas #5 ($1.2 \times 3.3 \, \mu m^2$) and #1 ($5.2 \times 2.3 \, \mu m^2$). The separation between generator and detector mesas is ~25 µm. Those are optimal configurations because (i) Larger generators allows larger total emission power (proportional to the area); (ii) Smaller detectors yield higher sensitivity (inversely proportional to the area); (iii) A significant separation between the generator and the detector minimizes unwanted response to direct heating[24] and non-equilibrium quasiparticle injection[33,34].

Figure 1c shows current–voltage ($I$–$V$) characteristics of the generator mesa #2. Above the Josephson critical current $I_c \sim 50$ µA, IJJ's start to switch one-by-one from the superconducting to the resistive state. In total there are about $N \simeq 250$ junctions. It is seen that all junctions have a similar $I_c$, indicating good

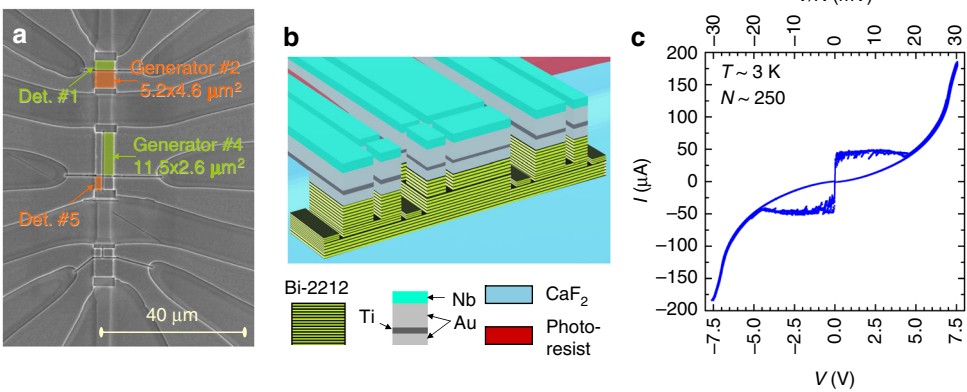

**Fig. 1** Sample characteristics. **a** Scanning electron microscopy image of the studied sample. The generator and the detector mesas are indicated. **b** A three-dimensional sketch of the sample. **c** Current–voltage characteristics of the generator mesa #2. The top axis shows voltage per junction. A kink at $V/N = 2\Delta/e \simeq 30$ mV represents the sum-gap singularity. It corresponds to ~15 THz upper Josephson frequency limit

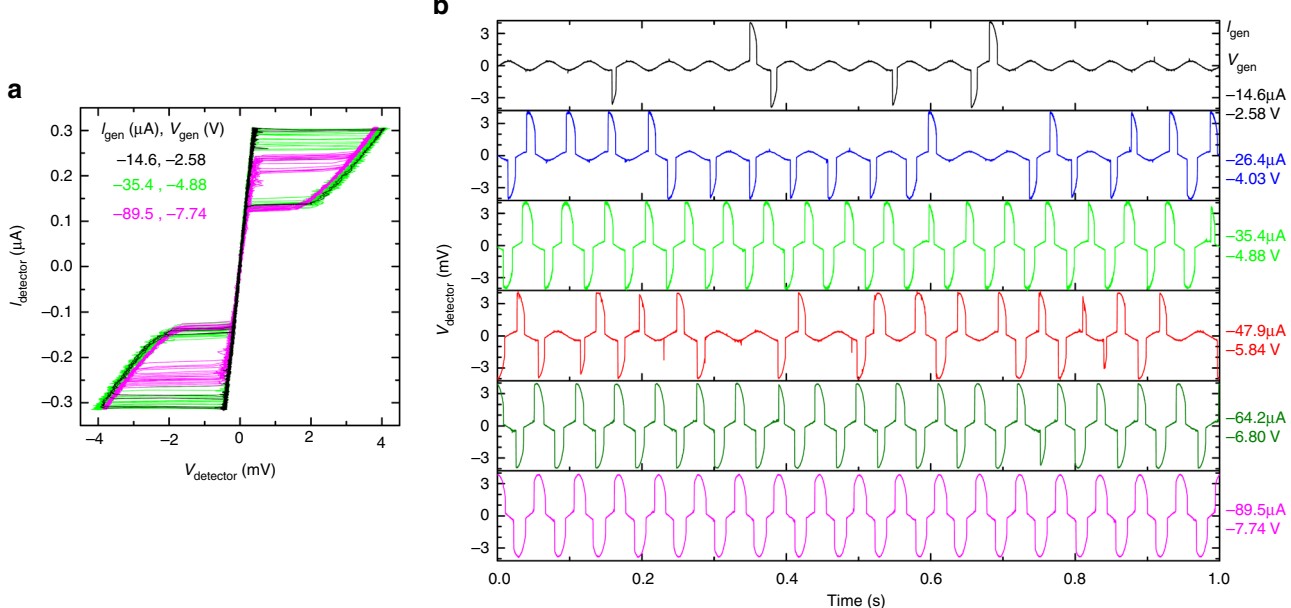

**Fig. 2** Operation of a switching current detector. **a** I–V characteristics of a top junction in the detector mesa #5 for different biases in the generator mesa #2. **b** Time dependence of the detector voltage. Different panels from top to bottom correspond to increasing negative bias in the generator. The number of voltage pulses reflect the switching probability. It is seen that the switching probability changes non-monotonously with the generator bias. The base temperature is $T \simeq 3\,\mathrm{K}$

uniformity of the mesa. The studied Bi-2212 crystal is strongly underdoped, $T_c \simeq 50\,\mathrm{K}$, $p \simeq 0.1$ (holes per Cu atom). This leads to a low critical current density[35] $J_c \sim 100\,\mathrm{A/cm^2}$ and allows reaching a high voltage ~10 V at a relatively small bias current $I \sim 200\,\mu\mathrm{A}$ and a modest dissipation power and self-heating. The top axis in Fig. 1c represents voltage per junction, normalized by the number of junctions. A sum-gap kink at $eV/N = 2\Delta \simeq 30\,\mathrm{meV}$ is seen. Although $\Delta$ is smaller than the equilibrium value for such crystals[14,35], it is still quite large. This indicates that gap suppression by self-heating and non-equilibrium quasiparticle injection is modest up to the highest bias.

**Switching current detector.** To detect the emission we use the top junction in the detector mesa as a switching current detector. Due to deterioration of the surface CuO layer, the top junction has a significantly lower critical current[14]. This allows independent biasing of the top junction while the rest of the detector mesa remains in the zero-voltage state. Figure 2a shows I–V characteristics of the top junction in the detector mesa #5 at different biases of the generator #2. The switching current detector operates in the following manner. A low-frequency AC current is applied to the junction with an amplitude slightly lower than the switching current from the superconducting to the resistive state in the absence of emission. Thus at zero bias in the generator (no emission) only few sporadic switchings occur, caused by thermal fluctuations. Incoming electromagnetic waves induce an additional high-frequency current in the detector junction and enhance the switching probability[36], see Methods section.

Figure 2b shows time-dependent readout of the detector for different dc-biases at the generator. At $V_{gen} = 0$ there are no switching events. With increasing $|V_{gen}|$ the switching events become more frequent. However, the switching probability increases non-monotonously with increasing $|V_{gen}|$. From Fig. 2b it is seen that it first goes up at $|V_{gen}| = 0{-}4.88\,\mathrm{V}$, but falls down at $5.84\,\mathrm{V}$ and then grows again. Such a non-monotonous response with respect to the total dissipation power at the generator precludes heating as the origin of the detected response.

The lack of heating is also obvious from independence of the quasiparticle resistance of the detector on the bias in the generator. This can be seen from coincidence of the resistive parts of the I–V's in Fig. 2a. The quasiparticle resistance has a strong temperature dependence[14] and is a sensitive probe of local temperature[24]. The lack of obvious heating and the non-monotonous variation of the switching current with increasing dissipation power at the generator, indicates that our detector senses induced electromagnetic currents, rather than local heating[24], or non-equilibrium effects[33,34,37]. The latter (unwanted) type of response appears at higher bias close or above the sum-gap voltage and limits the upper frequency range of detection (Supplementary Note 3).

Detected electromagnetic waves propagate through the open space and are guided to the detector mesa by antennae formed by contact electrodes. We are confident that the signal does not propagates through the base crystal. First of all, there is no dc-electrical crosstalk between generator and detector mesas. This is explicitly seen in Fig. 2a: application of a large bias into the generator does not lead to a shift of the detector I–V. The crosstalk is absent because the total bias current is small enough so that the base crystal remains in the superconducting state and the distance between the generator and detector is large enough to prevent penetration of non-equilibrium quasiparticles from the generator. At high frequencies the crosstalk through the base crystal should be even smaller because of the skin effect. Superconductors are very effective in screening electromagnetic waves. The skin depth in a superconductor even at low frequency is limited by the London penetration depth $\lambda < 200\,\mathrm{nm}$, which is ~100 times smaller than the separation between generator and detector mesas. The skin depth in superconductors rapidly (quadratically, as opposed to a square root dependence for normal metals) reduces with increasing frequency. At THz frequencies, comparable with the inverse scattering time, it is negligible. Since electromagnetic waves in the crystal decay exponentially with the characteristic length equal to the skin depth, we may safely neglect propagation of THz waves through the base crystal.

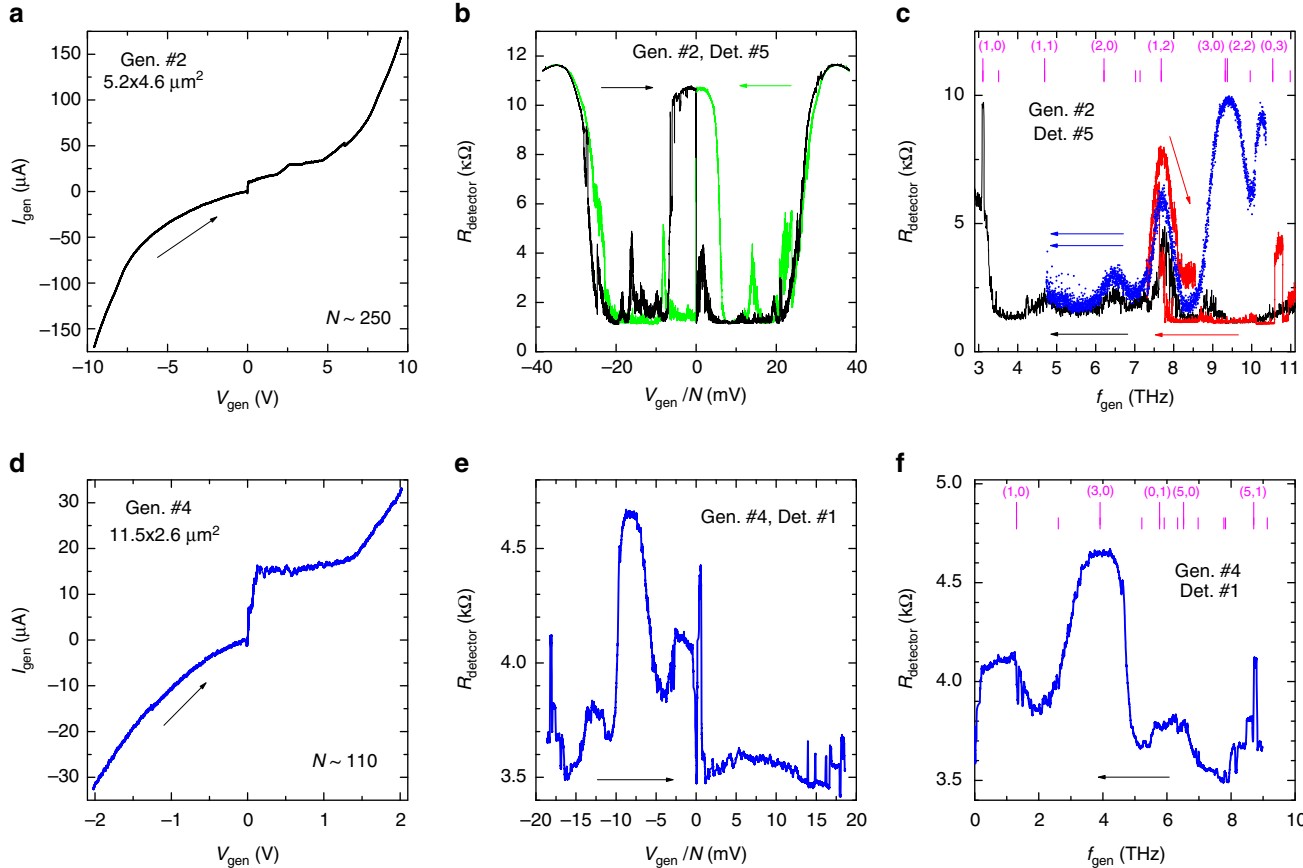

**Fig. 3** Generation detection experiment. For **a–c** generator #2, detector #5 and **d–f** generator #4, detector #1. The base temperature is $T = 3$ K. **a**, **d** I–V characteristics of generator mesas for bias sweep from negative to positive voltage. **b**, **e** Measured detector responses as a function of the detector voltage per junction. The green line in **b** represents detector response for a reverse bias sweep in the generator from positive to negative voltage. **c**, **f** Emission spectra: detector response vs. Josephson frequency for the falling parts of the generator I–V curves. Different colors in **c** represent different sweeps. Arrows indicate sweeping directions. Bars at the top indicate expected positions of primary cavity modes

**Generation detection experiment.** In Fig. 3 we show our main results for two measurement configurations: a–c generator #2, detector #5 and d–f generator #4, detector #1, see the schematics in Fig. 1a. Figure 3a, d show I–V characteristics of the generators for sweeps from a large negative to a large positive voltage. Mesa #4 was etched only half-way into the pedestal, see a sketch in Fig. 1b, and, therefore, contains only approximately half of the total number of junctions $N \sim 100$. This also limits the range of bias to $\pm 2$ V because at larger bias pedestal junctions start to switch into the resistive state, which complicates the analysis. The generator I–V's are asymmetric for falling and rising absolute values of $|V_{gen}|$. At the falling part, upon sweeping to zero, all junctions are in a similar resistive state. At the rising part, after passing through zero, junctions return into the superconducting state and with further increase of current above the critical current start sporadically one-by-one switching into the resistive state. This leads to appearance of multiple inner branches in the I–V's, Fig. 1c. Therefore, the same absolute value of voltage at falling and rising parts of the I–V may correspond to different numbers of active junctions. This is particularly well seen for the generator #4 in Fig. 3d.

Figure 3b, e show corresponding detector responses. They represent the lock-in resistance of the detector, which is essentially an integral of the absolute value of the detector voltage in Fig. 2b (Methods section). The higher is the detector resistance - the larger is the switching probability. For the detector #5, 100% switching probability corresponds to $R_{det} \simeq 12$ kΩ. It is seen that the detector response exhibits resonant peaks at

certain $V_{gen}$, confirming the non-thermal nature of the detected signal. This reflects the non-monotonous variation of the switching probability of the detector junction with varying the generator voltage, see Fig. 2b at $V_{gen} = -5.84$ V. The detector response also shows a clear asymmetry between falling and rising parts of the generator I–V. Most of the detected signal appears at the falling part when all junctions are active. It is much more difficult to get the emission at the rising part with inner branches. For example, in Fig. 3e there is practically no detected signal at $V_{gen} > 0$. In few cases, however, we did see some sporadic peaks at inner branches, as can be seen from Fig. 3b. Once such a peak is achieved, it remains stable for a fixed bias. But it is destroyed when extra junctions switch in or out of the resistive state. In Fig. 3b we also show detector response for the reverse sweep from +10 to −10 V (green). It is clear that it is approximately mirror symmetric with respect to the −10 to +10 V sweep. This demonstrates that it is not the sign of the generator current that matters, but the state of the junctions in the generator mesa. The emission signal is large at the falling part of $|V_{gen}|$ when all junctions are similarly active and significantly smaller at the rising part corresponding to sporadic switching of junctions (inner branches in the I–V's). The observed detection asymmetry is anticipated for electromagnetic wave emission from stacked Josephson junctions[26]. Falling part of the I–V with all junctions in the resistive (oscillating) state facilitates coherent superradiant enhancement of emission. On the other hand, raising part with inner junctions is usually associated with chaotic unsynchronized state of the stack with little emission[26].

In Fig. 3c, f we show the detected spectra as a function of the Josephson frequency, Eq. (1), obtained at the falling sides of the generator $I$–$V$'s with all junctions in the resistive state. Different colors in Fig. 3c represent different runs: from −10 to +10 V (black), from −10 to −2.5 and then to −4.4 V (red) and two subsequent sweeps in the smaller range from −5 to −2.5 V (blue). Vertical bars on top of Fig. 3c, f mark expected frequencies of strongly emitting in-phase geometrical resonances in a rectangular mesa with sizes $L_x \times L_y$ corresponding to those generators,

$$f(m, n) = \frac{c_1}{2} \sqrt{\frac{m^2}{L_x^2} + \frac{n^2}{L_y^2}}. \tag{2}$$

Here $c_1 \simeq 0.1c$ is the speed of in-phase electromagnetic waves, deduced from analysis of Fiske steps in similar small Bi-2212 mesas[25] ($c$- is the speed of light in vacuum). It is seen that the emission peaks correspond to certain cavity modes, indicated in brackets. Due to shape differences of the two generators the cavity modes are also considerably different. For example, the lowest (1, 0) cavity mode frequency in the generator #2 is more than two times higher than in the generator #4. Therefore, correlation of the emission maxima with shape-dependent cavity modes in the two generators (without fitting parameters) proves the essential role of in-phase geometrical resonances. This is in line with our expectations, because cavity modes both can help in synchronization of stacked junctions in the mesa[27,31], needed for coherent superradiant emission, and contribute to resonant enhancement of the emission power $\propto Q^2$[26,27]. An order of magnitude estimation of the maximum emission power yields ~1 μW, which corresponds to an encouraging DC to AC conversion efficacy (details of the estimation are described in Methods section).

## Discussion

The emission frequency range for both generators is very broad 3–11 THz for the generator #2 and 1–9 THz for the generator #4 (limited only by the bias range). Both numbers are record high for Josephson oscillators. We emphasize that emission takes place in the whole frequency range. However, optimal (maximal emission power) conditions are achieved at resonances when the Josephson frequency coincides with one of the cavity mode frequencies. Consequently, there are two ways of tuning the frequency: continuous, by changing the bias voltage; and discrete, by changing the mesa geometry. The continuous tuning is provided by the ac-Josephson effect, Eq. (1). That is, the frequency is simply tuned by changing the bias voltage. The discrete cavity modes spectrum, Eq. (2), can be tuned by changing (designing) the geometry (size and shape) of the mesa, as clearly follows from comparison of spectra in Fig. 3c, f. The reported here ten-fold frequency tunability 1–11 THz is by far more superior to that achieved in semiconductor quantum cascade lasers[5–8]. In contrast to the frequency comb method of tuning of such lasers[7], in which the overall emitted power is spread over the entire frequency comb, in our case practically all the emission power is concentrated in the (single) primary Josephson frequency with only a minor fraction going to higher harmonics[20,27]. Therefore, superconducting THz sources are superior in terms of tunability because the extreme broad range tunability of the primary frequency occurs without loss of radiation efficacy.

From Fig. 3c it is seen that the peak amplitudes depend on history and the maximum bias current, but their voltages are fixed. For example, two downward sweeps from high bias −10 to 0 V (black) and −10 to −2.5 V (red) produces slightly different set of peaks. While the two subsequent sweeps in a smaller bias range −5 to −2.5 V (blue) are fully reproducible but have yet another set of peaks. The observed history dependence is consistent with

vortex-antivortex (breather) mediated mechanism of emission[27]. Vortex-antivortex pairs in stacked junction may have different arrangements[32,38], which leads to history (initial condition) dependence of breather-type self-oscillations[27]. Vortices can be rearranged by high enough current[39], which can explain the observed rearrangement of emission peak amplitudes after applying high bias current. Importantly, however, that the peak frequencies are not changed.

The coincidence of the emission maxima with in-phase cavity modes points towards coherent superradiant emission from all the junctions in the generator mesa. The importance of coherency follows also from the discussed asymmetry of the emission with respect to the bias direction. At the falling part of the $I$–$V$ all the junctions are in the oscillating (resistive) state and it is at this side the emission takes place. Upon passing through $I_{gen} = 0$, junctions return into the stationary state $V = 0$ and with further increase of current sporadically switch into the resistive state. Here the stack is in the chaotic state with random phases between different junctions, which precludes superradiance[26]. Therefore, for raising $|V_{gen}|$ the emission is significantly smaller, despite a larger dc-power at the generator.

Finally, we want to compare our results with previous unsuccessful attempts to observe THz radiation from similar mesas (down to 0.1 pW of the absorbed power)[30]. The only difference is in the amount of junctions. Our mesas are high (contain more than hundred junctions), while unsuccessful attempts were all made for mesas with <100 junctions. The difference can not be explained by the $N^2$ dependence of superradiant emission power and rather suggests that there is a threshold number of junctions, required for emission. This is consistent with an observation that for the generator mesa #4 with $N \simeq 110$ only slightly above the threshold there is almost no emission at the rising part of the $I$–$V$ with inner branches when not all of the junctions are active, see Fig. 3e at positive $V_{gen}$. On the other hand, for the generator #2 with $N \simeq 250$ we could sometimes see emission even at the rising part of the $I$–$V$, see Fig. 3b.

The emission threshold is not present in the coupled sine-Gordon description of the mesas[25,26,32], which to the contrary predicts that stacks with small number of junctions are easier to synchronize and achieve emission[31]. On the other hand, a similar threshold behavior was observed for junction arrays[9]. The threshold appears when the synchronizing cavity mode is excited collectively by all junctions[40–42]. In this case the photon number in the cavity grows in a cascade manner[33,37] with the number of active junctions, like in a quantum cascade laser[5–8]. This leads to a laser-like threshold behavior either for the onset of global synchronization of the junction array[40] (classical case), or for the appearance of stimulated emission[37] (quantum case).

To conclude, we have demonstrated THz wave emission from small but high mesa structures made of a layered high-$T_c$ superconductor Bi-2212. The emission occurs at primary in-phase cavity modes in the mesa. We reported emission with a good efficacy in a broad 1–11 THz frequency range, which is record high for Josephson devices and is close to the limit set by the superconducting energy gap of the used cuprate. Our results suggest that there is a threshold number of junctions $N > 100$ needed for achieving the emission. This may indicate that the synchronizing cavity mode is pumped collectively by all the junctions, which leads to cascade amplification of the photon number, similar to that in quantum cascade lasers. This is important for development of tunable, compact, continuous-wave, narrow line-width Josephson THz sources. We anticipate that such superconducting oscillators with inherently small power dissipation would be preferable for application in high-frequency superconducting electronics[10,39].

## Methods

**Experimental**. Mesa structures are fabricated using standard micro/nano fabrication techniques. Measurements are performed in a cryogen-free cryostat with optical windows into the sample space. Details of sample fabrication, experimental setup and discussion of thermal respond of the detector are discussed in Supplementary Notes 1–3.

**Absolute sensitivity of the switching current detector**. The switching current detector allows direct measurement of the absorbed power $P_a$. The total instantaneous current in the detector is $I = I_{ac} + I_{THz}$. Here $I_{ac}$ is the low-frequency bias current and $I_{THz}$ is the induced high-frequency current. The induced THz-current leads to a premature switching to the resistive state[36]. Thus, the amplitude $I_{THz}$ is simply equal to the reduction of the switching current. From Fig. 2a it is seen that the maximum reduction is about $I_{THz} \simeq 0.1\,\mu A$. The absorbed power keeps the junction in the running state at the characteristic voltage $V_c \simeq 4\,mV$, Fig. 2a. It is equal to the work done in order to overcome the Josephson barrier[36]

$$\Delta U \simeq \frac{4\sqrt{2}}{3}\left(1 - \frac{I}{I_0}\right)^{3/2}\frac{\Phi_0 I_0}{2\pi}, \qquad (3)$$

every Josephson cycle $\Delta t = V_c/\Phi_0$. Thus, the absorbed power is

$$P_a \simeq \frac{2\sqrt{2}}{3\pi}\left(1 - \frac{I_s}{I_0}\right)^{3/2} I_0 V_c. \qquad (4)$$

Here $I_0$ is the fluctuation-free Josephson critical current and $I_s$ is the switching current suppressed by the THz-signal. Total suppression, $I_s = 0$, would require $P_a \simeq 0.5\,nW$ in this detector junction. The maximum absorbed power at the largest resonant peak in Fig. 3c is in the range of 0.1 nW.

**Calibration of the absorption efficiency using an external THz source**. We made a rough calibration of the THz absorption efficiency by similar mesas using an external THz source. The THz power with a frequency 1–1.2 THz from an external Backward-Wave Oscillator (BWO) was guided into the optical cryostat in a quasioptical manner using a set of high-density polyethylene lenses. The BWO THz beam is focused into a spot with the width at half maximum of 3 mm. We used an opto-acoustic Golay cell detector for measuring the transmission power for a fixed set of lenses and the detector either without the cryostat (through open space), or straight through the cryostat optical windows (without a sample). Optical windows were carefully shielded from optical pollution and room temperature radiation by Zitex filters placed at a 4 K thermal shield apertures. More experimental details can be found in ref.[30]. The detected power is reduced by a factor 6 upon transmission through the cryostat, primarily due to diffraction at relatively small apertures of our optical windows and also in part due to absorption by optical windows and filters. Since the BWO beam goes through two windows the reduction factor at the sample stage is 3 (two times less). The net emission power from BWO is ranging from 1 to 0.5 mW in the frequency range from 1 to 1.2 THz. This yields ~0.2–0.3 mW at the sample with the power density of 1–2 mW/cm². With those parameters, a detector mesa with a similar geometry, but higher critical current of the top junction $I_0 = 100\,\mu A$ and the absorption sensitivity of few nW, barely detects the BWO signal. Thus the absorption detection efficiency of the detector, i.e., the ratio of the absorbed to the input THz power is low $\leq 10^{-4}$. The low absorption efficiency is due to a small area of the detector with respect to the spot size and absence of specially matched antenna for picking up the THz radiation. We note that this is a rough, order-of-magnitude estimation of the absorption efficiency by the detector.

**Estimation of the emitted power and the emission efficacy**. Knowing the absorbed power (maximum 0.1 nW) and the absorption efficiency ($\leq 10^{-4}$), we can estimate the maximum emitted power ~1 µW, which corresponds to a few percent of the total dc-power, $P_{dc} = IV$, in the generator mesa. Such an efficacy (ratio of the emitted to the consumed power) is comparable to that for large Bi-2212 mesas[11,12,16,17,20]. We note that this is only an order of magnitude estimation due to roughness of calibration of the absorption efficiency. The maximum theoretically achievable efficacy is 50%[26]. Properly designed single junction Flux-Flow oscillators may have efficacy of $\leq 10\%$[10]. However, in those devices special care is taken for impedance matching, needed for effective emission from the junction. Furthermore, the electromagnetic signal in that case is transferred via a transmission line, rather than open space. Therefore efficacy of few percent, without proper antenna design, is quite encouraging.

**Data availability**. All data are available from the corresponding author upon reasonable request.

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

## Acknowledgements

The work was supported by the Swedish Foundation for International Cooperation in Research and Higher Education Grant No. IG2013-5453 and the Swedish Research Council Grant No. 621-2014-4314. We are grateful to A.B. Kulakov for providing Bi-2212 single crystals. Technical support from the Core Facility in Nanotechnology at Stockholm University is gratefully acknowledged.

## Author contributions

E.A.B. made the sample and performed the experiment with input from V.M.K.; V.M.K. conceived the project and wrote the manuscript with input from E.A.B.

## Additional information

**Competing interests:** The authors declare no competing financial interests.

