## [Peer Review File · Nature Communications]

Reviewers' comments:

Reviewer #1 (Remarks to the Author):

The manuscript "Record high frequency 3-11 THz Josephson emission from small-but-high Bi₂Sr₂CaCu₂O_{8+δ} mesa structures" by E. A. Borodianskyi and V. M. Krasnov concerns emission from mesa structures of Josephson junctions at THz frequencies higher than what has been reported until now. The main advance of the structures used in this manuscript is that the number of junctions stacked in the small Bi-2212 mesa has been significantly increased to contain 250 junctions which is more than twice as much than what has typically been used in previous reports. The THz radiation at higher frequencies is identified to be higher order in-phase cavity modes observed as peaks between 3 and 11 THz. While the THz emission is detected by a similar but smaller-area mesa on the same sample, experiments are performed where the bias of the generator and detector mesa are swept individually in order to quantitatively estimate the detector sensitivity and emission power, and exclude any thermal effects.

Although I do acknowledge the advance of the increased number of Josephson junctions leading to THz emission at higher frequencies than previously reported, the manuscript is only presenting an incremental improvement for frequency scaling and hence I do not find it suitable for publication in Nature Communications. I believe that the manuscript is more appropriate for a technical or THz/superconductor specific journal.

Below I have listed a few suggestions for a revised manuscript:

- While the introduction gives a good overview of Josephson junctions, the feasibility of Josephson junctions as THz emitters is poorly described. In today's THz research "the gap" is more or less covered since air plasma sources and detectors can provide tens of THz bandwidth [Thomson, M. D., Blank, V. & Roskos, H. G. Terahertz white-light pulses from an air plasma photo-induced by incommensurate two-color optical fields. *Opt. Express* 18, 23173-23182 (2010)], certain nonlinear crystals also emit beyond 3 THz ([Valverde-Chávez, D. A. & Cooke, D. G. Multi-Cycle Terahertz Emission from β -Barium Borate. *J. Infrared Millim. Terahertz Waves* 38, 96-103 (2016)] and [Shalaby, M. & Hauri, C. P. Demonstration of a low-frequency three-dimensional terahertz bullet with extreme brightness. *Nat. Commun.* 6, 5976 (2015)]) and even commercial systems based on photoconductive antennas approach bandwidths of 6 THz (see for example Toptica's THz-TDS systems). Nevertheless, all of the above techniques are based on advanced laser systems while the Josephson junctions don't require optical pumping which I think could be stressed more.

- The introduction doesn't describe the content of the manuscript which makes the story seem unclear.

- In the abstract it is stated that the authors “could achieve tunable THz emission” which is not evident from the data presented manuscript. Despite that different spectra are obtained depending on the direction of the bias sweep this seem neither to be controllable or actually to give any clear frequency shift of the individual peaks. If the tunability is meant as a possibility to tune the frequency of the cavity modes by changing L_x and/or L_y this should be experimentally verified.
- For Fig. 1: instead of showing the same data twice with a different x-axis, the both x-axes could be shown in the same figure below and above the data, respectively. It would be helpful if a sketch of the sample from side/perspective view were shown in Fig. 1 as well together with the SEM image.
- The discussion of the results in the manuscript is mostly concerned with excluding any thermal effects to be the source of the detected current which understandably is an important issue. While I agree that the non-monotonous trend of Fig. 2 (a) is unlikely to be a thermal effect I am somewhat puzzled with the behavior when the generator bias is -5.84 V. This should be explained in greater detail.
- While the fact that the experiment is performed with an emitter and detector mesa on the same sample unit surely is neat, I also believe that it gives rise to some concerns about the estimated optical power and sensitivity which are essential points of the manuscript. It seems that the authors have verified that the two mesas are electrically decoupled, however, the optical coupling between the emitter and detector is not described. In the geometric configuration of the sample I believe this should be poor, especially since the detector is smaller than the generator. The coupling could be estimated by finite element method simulations and compared with Josephson junctions on a silicon lens for coupling to free-space, see [Ji, M. et al. Bi2Sr2CaCu2O8 intrinsic Josephson junction stacks with improved cooling: Coherent emission above 1 THz. Appl. Phys. Lett. 105, 122602 (2014)] and [Kakeya, I. & Wang, H. Terahertz-wave emission from Bi2212 intrinsic Josephson junctions: a review on recent progress. Supercon. Sci. Technol. 29, 073001 (2016).].
- I question whether the estimate of the absorption efficiency is reasonable since the used BWO THz source emits at 1.1 THz while the frequencies of the Josephson detector are shown between 3-11 THz. For most THz detectors it is quite unlikely that the response is similar at 1 THz and at 11 THz so this should be verified. Considering the spatial geometry rather than the frequency response I believe that it is also very likely that the optical coupling of the BWO is just too tricky and not comparable at all to the situation where the emission is detected by the other Josephson junction.
- In relation to the question above: if the emitted power really is on the order of 1 μW it should be possible to detect it with the Golay cell if collecting with an off-axis parabolic mirror (and refocusing on to the Golay cell with another off-axis parabolic mirror). If the Golay cell is sensitive enough for this experiment I believe this is a better way to estimate the emission power and the sensitivity of the Josephson detector.
- The discussion of the asymmetry of Fig. 3 (b) leading to the “history” effect of the junction is good, but it also leads to questions about if the data is really reproducible. This should be discussed and maybe even included as a scan from +10 to 0 V.
- The reference [Wang, H. B. et al. Coherent terahertz emission of intrinsic Josephson junction stacks in the hot spot regime [corrected]. Phys. Rev. Lett. 105, 057002 (2010).] is not mentioned in the

paper. Although the earlier work is mentioned in Ref. 9 is included the PRL above is relevant since the THz spectra are included.

Reviewer #2 (Remarks to the Author):

The authors investigated Terahertz emission properties of mesa structures patterned on strongly underdoped Bi-2221 single crystals. The mesa structures form stacks of intrinsic Josephson junctions (IJJs). Previous experiments focused on very large and moderately underdoped stacks with in-plane dimensions in the 100 μm range, consisting of 500 or more IJJs. By contrast the present study uses small mesas with in-plane sizes of around 5 μm and a junction number of about 250. These stacks are much less prone to Joule heating. At the same time they consist of much more IJJs than small stacks used previously. The authors detect THz emission by using a detector stack fabricated on the same Bi-2212 single crystal, studying its switching probability from the zero-voltage state to the resistive state. There is good indication that the emitter stack emits at frequencies between 3 and 11 THz, which is much higher than the emission frequencies of large stacks. The results are novel and highly interesting for other researchers in the field. The subject is suitable for publication in Nature Communications.

I have a number of questions and suggestions.

(1) The title is good for experts, however perhaps not for a general audience. This particularly regards "small-but-high mesa structures".

(2) Abstract, "approaching a theoretical limit for Josephson oscillators": This is true for cuprates but not for all superconductors, please rephrase.

(3) first paragraph, line 5: e and h are undefined.

(4) first paragraph, lines 14 and 15: "large Bi-2212 mess". Define "large" and correct "mess" to "mesas".

(5) second column, second paragraph: "In what follows we will show results for a configuration...": Please also comment on the other mesas/configurations. Were similar results obtained here?

(6) Use of the detector junction as a switching detector: The switches are actually not frequency selective. Was there a Shapiro-step-like response on the resistive branch of the current voltage characteristic? Even if the irradiated THz power is too low I would expect some small response when the emission frequency and the Josephson frequency of the detector match. Otherwise the authors should consider to weaken the claims "we have demonstrated THz emission" (conclusions) and "we could achieve tunable THz emission" (abstract), using phrases like "we have found strong evidence for..." etc.

(7) What happens if the generator is biased on one of its inner branches? Is there a response of the detector and, if so, how does it change with the number of detector IJJs in the resistive state?

(8) Supplement, Experimental Setup, second paragraph, size of mesas: replace μ^2 by μ_m^2 .

Reviewer #3 (Remarks to the Author):

This paper reports the new experimental evidence that electromagnetic waves at a THz frequency region between 3 and 11 THz was detected from the mesa structure made from intrinsic Josephson junction of Bi2212 crystal with the number of Josephson junctions of $N=250$. In spite of a potential THz emission up to a few tens of THz because of the large superconducting gap, of the order of $\Delta \approx 30$ meV (corresponding to 15 THz), no such an experiment above 2.4 THz has been reported. In this regard, this paper made one step further, exploring higher THz frequency region using intrinsic Josephson junction device. This result certainly deserve and is worthwhile publishing in the Nature communications.

Before publication, however, the authors should consider and comment to one important point on why in-line self detection method using intrinsic Josephson junction mesa structure made of Bi2212 crystal located very near to the THz emitter is used for the THz detection. The ac-Josephson current must be there, certainly inside the mesa, and probably in the vicinity of the mesa, too, because of the leakage of the ac current from the mesa itself, since there is no doubt about the ac Josephson effect working properly. However, there may be some concern on the emission of THz waves to the vacuum space. Therefore, in the present experiment, hopefully, the THz detector should be set separately and independently from the emitter to ensure that the THz radiation is actually emitted, propagated through vacuum and reached at the detector. In order to make this experimental report more confident from skepticism and more invaluable the authors needs to add explanation on the detection technique.

After revision concerning this point, the referee agree with publication to Nature communications.

Reply to reviewers

1. Reviewer #1 writes:

“The manuscript “Record high frequency 3-11 THz Josephson emission from small-but-high Bi₂Sr₂CaCu₂O_{8+δ} mesa structures” by E. A. Borodianskyi and V. M. Krasnov concerns emission from mesa structures of Josephson junctions at THz frequencies higher than what has been reported until now. The main advance of the structures used in this manuscript is that the number of junctions stacked in the small Bi-2212 mesa has been significantly increased to contain 250 junctions which is more than twice as much than what has typically been used in previous reports. The THz radiation at higher frequencies is identified to be higher order in-phase cavity modes observed as peaks between 3 and 11 THz. While the THz emission is detected by a similar but smaller-area mesa on the same sample, experiments are performed where the bias of the generator and detector mesa are swept individually in order to quantitatively estimate the detector sensitivity and emission power, and exclude any thermal effects.

Although I do acknowledge the advance of the increased number of Josephson junctions leading to THz emission at higher frequencies than previously reported, the manuscript is only presenting an incremental improvement for frequency scaling and hence I do not find it suitable for publication in Nature Communications. I believe that the manuscript is more appropriate for a technical or THz/superconductor specific journal.”

Reply-1

In this work we report on practically a ten-fold increment of the primary emission frequency from a Josephson oscillator (we are not discussing higher harmonics, as in some previous works). We approached a theoretical limit for such a superconducting oscillator. This is also the first successful report made on small-but-high mesa structures, which has not been done before. This is new. The unique property of our oscillator is that a single device can tunably cover practically the full THz range from below 1 THz (we added a new data to demonstrate this) to more than 10 THz at a primary frequency. For comparison, semiconducting Quantum Cascade Lasers in a frequency-comb mode has only a factor-two frequency span [we added discussion and new references in the modified version]. Therefore, the achieved ten-fold frequency span is unique for any laser-like Narrow Line-width, Continuous-Wave (CW) oscillator. Therefore, we have difficulty to agree with the classification of our results as an “incremental improvement”. Our results have to be compared with earlier efforts on CW THz emission from large mesas, which after initial success came to stagnation when it comes to increment of the primary emission frequency. We hope that our discovery, demonstration of a possibility of the broad-range but narrow line THz emission from small-but-high mesas, will boost the activity in this area. In the modified version of the manuscript we added new data and modified the text, trying to emphasize the novelty and significance of our results. We also tried to meet the critics of the reviewer and answered on the questions in the text and in the reply below. We hope that this will remove a potential misunderstanding and that the reviewer will be able to better appreciate our work in its modified version.

2. Reviewer #1 writes:

“Below I have listed a few suggestions for a revised manuscript:

- While the introduction gives a good overview of Josephson junctions, the feasibility of Josephson junctions as THz emitters is poorly described. In today’s THz research “the gap” is

more or less covered since air plasma sources and detectors can provide tens of THz bandwidth [Thomson, M. D., Blank, V. & Roskos, H. G. Terahertz white-light pulses from an air plasma photo-induced by incommensurate two-color optical fields. *Opt. Express* 18, 23173-23182 (2010)], certain nonlinear crystals also emit beyond 3 THz ([Valverde-Chávez, D. A. & Cooke, D. G. Multi-Cycle Terahertz Emission from β -Barium Borate. *J. Infrared Millim. Terahertz Waves* 38, 96-103 (2016)] and [Shalaby, M. & Hauri, C. P. Demonstration of a low-frequency three-dimensional terahertz bullet with extreme brightness. *Nat. Commun.* 6, 5976 (2015)]) and even commercial systems based on photoconductive antennas approach bandwidths of 6 THz (see for example Toptica's THz-TDS systems). Nevertheless, all of the above techniques are based on advanced laser systems while the Josephson junctions don't require optical pumping which I think could be stressed more.

- The introduction doesn't describe the content of the manuscript which makes the story seem unclear.”

Reply-2

It is not only optical pumping, which is relevant here. As we mentioned in the introductory part, we are aiming to develop coherent, narrow-line (laser-like), continuous wave THz oscillators, which are also compact and tunable. Therefore, we are not competing with broad-band pulsed techniques, mentioned by the reviewer. Rather the nearest competitor is a quantum cascade laser. We added corresponding references and discussion in the modified version. When it comes to potential applications – our opinion is that Josephson oscillators will not compete with semiconductor or gas-lasers, but rather be useful as local oscillators for low-temperature applications. Especially in superconducting applications when the total power dissipation is crucial and the total power is not (microwatt is enough). Today such applications are clock oscillators in superconducting digital devices (RSFQ data acquisition and computer components) and local oscillators in heterodyne mixers/detectors. We added corresponding clarifications in the very end of the modified manuscript.

Following reviewer's advice, we changed the introductory part (1st paragraph). In which instead of the general THz-gap term we now discuss more carefully different THz sources. We added discussion of pulsed sources, including references mentioned by the reviewer, and specified that our goal is to develop coherent, narrow-line (laser-like) continuous wave, compact and tunable frequency THz oscillators. We also added references to frequency-comb Quantum Cascade Lasers as the nearest competitor.

3. Reviewer #1 writes:

“• In the abstract it is stated that the authors “could achieve tunable THz emission” which is not evident from the data presented manuscript. Despite that different spectra are obtained depending on the direction of the bias sweep this seem neither to be controllable or actually to give any clear frequency shift of the individual peaks. If the tunability is meant as a possibility to tune the frequency of the cavity modes by changing L_x and/or L_y this should be experimentally verified.”

Reply-3:

This is indeed an important issue. Josephson emission is a two-step process: First, Josephson oscillations are excited in voltage-driven junctions due to the ac-Josephson effect. Second, they pump one of the high-Q cavity modes in the mesa, which then emits the amplified electromagnetic wave out of the mesa. As we discussed in the introduction, the role of the cavity mode is essential both for synchronization of stacked junctions in the mesa, needed for

coherent superradiant emission and for enhancement of the emission power. Consequently, there are two ways of tuning the frequency. The continuous tuning is provided by the ac-Josephson effect: $f=2eV/h$. I.e., the frequency is simply tuned by changing the bias voltage. However, optimal (maximal emission power) conditions are achieved at resonances when the Josephson frequency coincides with one of the cavity mode frequencies, as shown in Figs. 3 (c) and (f). The cavity modes are not continuous, but discrete. In the modified version we marked all cavity modes (not only active ones) in order to show all potential resonant frequencies for a given mesa. As can be seen from Figs. 3 (c) and (f), the frequency is tunable in the full range, but the emission power (efficacy) is varying following the discreteness of the cavity-mode spectra of the mesa. The reviewer correctly anticipated that the cavity mode spectra can be tuned by changing the geometry of the mesa. In the modified version of the manuscript we added new data for emission from an elongated mesa #4. The lowest frequency was shifted correspondingly to ~ 1 THz. We thank the reviewer for the constructive suggestion, which helped to improve the manuscript and strengthen our conclusions.

4. Reviewer #1 writes:

“• For Fig. 1: instead of showing the same data twice with a different x-axis, the both x-axes could be shown in the same figure below and above the data, respectively. It would be helpful if a sketch of the sample from side/perspective view were shown in Fig. 1 as well together with the SEM image. “

Reply-4:

We followed reviewer's advice.

5. Reviewer #1 writes:

“• The discussion of the results in the manuscript is mostly concerned with excluding any thermal effects to be the source of the detected current which understandably is an important issue. While I agree that the non-monotonous trend of Fig. 2 (a) is unlikely to be a thermal effect I am somewhat puzzled with the behavior when the generator bias is -5.84 V. This should be explained in greater detail.”

Reply-5:

The downturn of the emission power at this bias occurs because the mesa goes out of the optimal emission condition. The Josephson frequency falls in-between two resonances. A clarification is added in the modified version.

6. Reviewer #1 writes:

“• While the fact that the experiment is performed with an emitter and detector mesa on the same sample unit surely is neat, I also believe that it gives rise to some concerns about the estimated optical power and sensitivity which are essential points of the manuscript. It seems that the authors have verified that the two mesas are electrically decoupled, however, the optical coupling between the emitter and detector is not described. In the geometric configuration of the sample I believe this should be poor, especially since the detector is smaller than the generator. The coupling could be estimated by finite element method simulations and compared with Josephson junctions on a silicon lens for coupling to free-space, see [Ji, M. et al. Bi₂Sr₂CaCu₂O₈ intrinsic Josephson junction stacks with improved cooling: Coherent emission above 1 THz. Appl. Phys. Lett. 105, 122602 (2014)] and [Takeya, I. & Wang, H. Terahertz-wave emission from Bi₂212 intrinsic Josephson junctions: a review on recent progress. Supercon. Sci. Technol. 29, 073001 (2016).].”

Reply-6:

Focusing of the emission would of course be useful. However, it is not the detector mesa that picks up the radiation. It is too small and even focusing would not help. Rather those are electrodes of the detector that act as antennas and pick up the signal. The corresponding area is fairly large. In a similar manner, electrode antennae of the generator emit the radiation. We added the corresponding clarification in the modified version.

We thank the referee for the relevant reference [Kakeya & Wang] and added it to our reference list.

7. Reviewer #1 writes:

“• I question whether the estimate of the absorption efficiency is reasonable since the used BWO THz source emits at 1.1 THz while the frequencies of the Josephson detector are shown between 3-11 THz. For most THz detectors it is quite unlikely that the response is similar at 1 THz and at 11 THz so this should be verified. Considering the spatial geometry rather than the frequency response I believe that it is also very likely that the optical coupling of the BWO is just too tricky and not comparable at all to the situation where the emission is detected by the other Josephson junction.”

Reply-7

In reply to this question we would like to provide several arguments:

As discussed above, the coupling occurs between the incoming electromagnetic wave and the antenna formed by the detector mesa electrodes. In this respect the origin of the radiation (BWO or generator) doesn't matter. What matters is the frequency dependence of the antenna+mesa impedance. In the new version of manuscript we added data for another generator/detector configuration, see Fig. 3 (f). Here we show response from zero frequency and the first maximum corresponds to a similar range of frequencies as in BWO. Furthermore, due to geometrical differences of the two generator mesas the lowest (1,0) cavity mode frequency is more than 2 times lower in this case. While electrodes at the detector mesas are similar at the wavelength scale. Therefore, we do not expect any significant difference in the frequency dependencies of antennae impedances of the two presented generator/detector experiments from Figs. 3 (c) and (f). From comparison of Figs. 3 (c) and (f) it is clear that there are no common features in those responses that could be attributed to the antennae. E.g. at 4THz there is a minimum in Fig. 3 (c) and a maximum in Fig. 3 (f). To the contrary, all the maxima in the two spectra are well correlated solely with cavity modes in the generators (marked by vertical bars). From those observations we may conclude that the detector antennae have fairly flat frequency responses in the corresponding frequency range. This substantiates the performed calibration.

Nevertheless, we agree that this is not a particularly accurate calibration. Therefore, we only call it “order of magnitude estimation”. This means that it could be 0.1 instead of 1 μW , but it could also be larger. We suppose that this should not be confusing as soon as it is clearly stated. To minimize the chance of confusion, in the modified version of the manuscript we explicitly mention throughout the manuscript that this is a rough order-of-magnitude estimation. For example, in the main text it now reads “An order of magnitude estimation of the maximum emitted power yields $\sim 1 \mu\text{W}$ ”, which is softer than the previous: “A conservative estimation of the maximum emitted power is of the order of a μW ”

8. Reviewer #1 writes:

“• In relation to the question above: if the emitted power really is on the order of 1 μW it should be possible to detect it with the Golay cell if collecting with an off-axis parabolic

mirror (and refocusing on to the Golay cell with another off-axis parabolic mirror). If the Golay cell is sensitive enough for this experiment I believe this is a better way to estimate the emission power and the sensitivity of the Josephson detector.”

Reply-8.

We definitely agree, but we can not do such an experiment right away because of the low detection efficiency of our present system with a Golay cell. Here is a simple estimation of the emission power outside our cryostat (assuming a simple spherical distribution of the power density): $P_{out} = a \cdot P_{gen} / (4 \cdot \pi \cdot L^2) \cdot (\pi \cdot D^2 / 4)$

Here $L=20$ cm is the distance from the sample to the optical window, $D=1$ cm is the diameter of the optical window, and $a \sim 0.1$ is the realistic loss factor (~ 0.3 diffraction and absorption losses in the cryostat, on the way from the sample to the optical window and ~ 0.3 in lenses towards the Golay cell). For the total generated power of $P_{gen}=1$ μ W this yields $P_{out} \sim 16$ pW, which is beyond the resolution of the Golay cell. As reviewer suggested, one would need to introduce an in-situ mirror or a lens to increase the effectiveness, which however unlikely to exceed a 1% efficiency of collection in any case (typical for measurements made on large mesas). Anyway, we can not perform such a measurement at a moment due to a very restricted sample space in our cryostat. We are considering ways to perform more “independent” measurements in the future. However, our data undoubtedly demonstrates that there is THz emission with a significant power. We can conclude this from comparison with numerous previous (negative and therefore largely unpublished) experiments with a similar technique on similar mesas will lesser amount of junctions.

9. Reviewer #1 writes:

- The discussion of the asymmetry of Fig. 3 (b) leading to the “history” effect of the junction is good, but it also leads to questions about if the data is really reproducible. This should be discussed and maybe even included as a scan from +10 to 0 V.

Reply-9.

We added a required scan from +10 to -10 V to the modified Fig. 3 (b). As discussed, there is no full reproducibility after going to very large bias. As we explained in the manuscript, this can be due to rearrangement of vortices in the generator mesa. However, in a narrower range there is a good reproducibility. This is demonstrated by the blue symbols, which represent two subsequent down-up sweeps. We also added a discussion of asymmetry of emission at falling and raising parts of the I-V.

10. Reviewer #1 writes:

- The reference [Wang, H. B. et al. Coherent terahertz emission of intrinsic Josephson junction stacks in the hot spot regime [corrected]. Phys. Rev. Lett. 105, 057002 (2010).] is not mentioned in the paper. Although the earlier work is mentioned in Ref. 9 is included the PRL above is relevant since the THz spectra are included.”

Reply-10.

We followed reviewer’s advice and added the corresponding reference.

Finally we want to thank the reviewer for valuable remarks that helped to improve the manuscript. We hope that corresponding modifications, additional data and our clarifications will convince the reviewer about novelty and significance of the presented report on occurrence of record-high frequency Josephson emission from small-but-high Bi-2212 mesa structures.

Reviewer #2 (Remarks to the Author):

1. Reviewer #2 writes:

The authors investigated Terahertz emission properties of mesa structures patterned on strongly underdoped Bi-2221 single crystals. The mesa structures form stacks of intrinsic Josephson junctions (IJJs). Previous experiments focused on very large and moderately underdoped stacks with in-plane dimensions in the 100 μm range, consisting of 500 or more IJJs. By contrast the present study uses small mesas with in-plane sizes of around 5 μm and a junction number of about 250. These stacks are much less prone to Joule heating. At the same time they consist of much more IJJs than small stacks used previously. The authors detect THz emission by using a detector stack fabricated on the same Bi-2212 single crystal, studying its switching probability from the zero-voltage state to the resistive state. There is good indication that the emitter stack emits at frequencies between 3 and 11 THz, which is much higher than the emission frequencies of large stacks. The results are novel and highly interesting for other researchers in the field. The subject is suitable for publication in Nature Communications.

Reply-1

We want to thank the reviewer for appreciation of our work and valuable remarks. In the modified version we made changes following reviewer's suggestions, as described in the replies below.

2. Reviewer #2 writes:

I have a number of questions and suggestions.

(1) The title is good for experts, however perhaps not for a general audience. This particularly regards "small-but-high mesa structures".

Reply-2

Following referees suggestions we modified the title.

2. Reviewer #2 writes:

(2) Abstract, "approaching a theoretical limit for Josephson oscillators": This is true for cuprates but not for all superconductors, please rephrase.

Reply-2

We specified that it concerns this particular high-Tc cuprate.

3. Reviewer #2 writes:

(3) first paragraph, line 5: e and h are undefined.

Reply-3

Fixed.

4. Reviewer #2 writes:

(4) first paragraph, lines 14 and 15: "large Bi-2212 mess". Define "large" and correct "mess" to "mesas".

Reply-3
Fixed.

5. Reviewer #2 writes:

(5) second column, second paragraph: "In what follows we will show results for a configuration...": Please also comment on the other mesas/configurations. Were similar results obtained here?

Reply-5

In the modified version of the manuscript we added data for a different generator/detector configuration, new Figs. 3 (d-f). They strengthen our conclusion concerning cavity mode origin of the emission peaks because that mesa had a considerably different geometry with significantly (more than twice) different resonant frequencies. It also shifted the lower range of emission to ~ 1 THz. We have modified the frequency span in the title and the text correspondingly (1-11 THz).

6. Reviewer #2 writes:

(6) Use of the detector junction as a switching detector: The switches are actually not frequency selective. Was there a Shapiro-step-like response on the resistive branch of the current voltage characteristic? Even if the irradiated THz power is too low I would expect some small response when the emission frequency and the Josephson frequency of the detector match. Otherwise the authors should consider to weaken the claims "we have demonstrated THz emission" (conclusions) and "we could achieve tunable THz emission" (abstract), using phrases like "we have found strong evidence for..." etc.

Reply-6

The sensitivity of the switching current detector on a single top junction is very much better than the Shapiro-step response of the whole detector mesa with all IJJs in the resistive state. Indeed, as the total absorbed power is only $P_a = 0.1$ nW, the amplitude of the Shapiro step is less than $dI = P_a/V$. With the V of the Shapiro step in the detector \sim the same as V generator = few Volt, dI is < 0.1 nA, which is beyond the resolution of simple dc-measurements. Moreover, the power dependence of the 1st Shapiro step is such that it has a maximum when the critical current is suppressed to zero. From Fig. 2 (a) it is seen that we can not suppress I_c even in the top junction. For the rest of "bulk" IJJs in the detector mesas the I_c is much larger and is barely suppressed at all. This further reduces the Shapiro step response in bulk IJJs to an unmeasurable value. For comparison we may say that we have never been able to observe Shapiro step response even using the BWO oscillator with ~ 0.3 mW focused power at the chip. The switching current detector is much more sensitive because it operates at zero voltage. Thus, unmeasurably small amplitude of Shapiro steps is not surprising. In the modified version we have strengthen our conclusions by adding new data. Nevertheless, we did follow reviewer's recommendation and softened the statement in the abstract to "strong evidence".

7. Reviewer #2 writes:

(7) What happens if the generator is biased on one of its inner branches? Is there a response of the detector and, if so, how does it change with the number of detector IJJs in the resistive state?

Reply-7

As we mentioned, there is a clear asymmetry between rising (inner branches) and falling (common quasiparticle branch) parts of the I-V, as shown in Figs. 3 (b) and (e). Generally, it is much more difficult to get the emission at the rising part because here junctions switch sporadically and the stack is in a chaotic unsynchronized state. For example, in Fig. 3 (e) there is practically no detected signal at the rising part of the I-V. In few cases, however, we did see some sporadic peaks at inner branches, as can be seen from Fig. 3(b). Once such a peak is achieved, it remains stable for a fixed bias. But it is destroyed when extra junctions switch in or out of the resistive state, destroying the synchronization. We added the corresponding discussion to the text.

8. Reviewer #2 writes:

(8) Supplement, Experimental Setup, second paragraph, size of mesas: replace μ^2 by μ_m^2 .

Reply-8

Fixed (moved to the main text).

Reviewer #3 (Remarks to the Author):

1. Reviewer #3 writes:

“This paper reports the new experimental evidence that electromagnetic waves at a THz frequency region between 3 and 11 THz was detected from the mesa structure made from intrinsic Josephson junction of Bi2212 crystal with the number of Josephson junctions of $N=250$. In spite of a potential THz emission up to a few tens of THz because of the large superconducting gap, of the order of $\Delta \approx 30$ meV (corresponding to 15 THz), no such an experiment above 2.4 THz has been reported. In this regard, this paper made one step further, exploring higher THz frequency region using intrinsic Josephson junction device. This result certainly deserve and is worthwhile publishing in the Nature communications.”

Reply-1

We want to thank the reviewer for appreciation of our work and valuable remarks. In the modified version we made changes following reviewer’s suggestions, as described in the replies below.

2. Reviewer #3 writes:

Before publication, however, the authors should consider and comment to one important point on why in-line self detection method using intrinsic Josephson junction mesa structure made of Bi2212 crystal located very near to the THz emitter is used for the THz detection. The ac-Josephson current must be there, certainly inside the mesa, and probably in the vicinity of the mesa, too, because of the leakage of the ac current from the mesa itself, since there is no doubt about the ac Josephson effect working properly. However, there may be some concern on the emission of THz waves to the vacuum space. Therefore, in the present experiment, hopefully, the THz detector should be set separately and independently from the emitter to ensure that the THz radiation is actually emitted, propagated through vacuum and reached at the detector. In order to make this experimental report more confident from skepticism and more invaluable the authors needs to add explanation on the detection technique.

After revision concerning this point, the referee agree with publication to Nature communications.

Reply-2

This is a relevant question. The main reasons for using an in-situ detector are: Outstanding sensitivity (pW), small losses on the way, absolute calibration (of absorbed power). We note that the switching current detector is a well known type of the superconducting detector. We are confident that the detected signal propagates through vacuum, not crystal. We specifically demonstrated no visible electrical cross-talk with the generator mesa at dc-current. At THz frequency the cross talk is even smaller because of the skin effect in the superconducting electrodes (base crystal). Superconductors are very effective in screening electromagnetic waves. The skin-depth in a superconductor even at low frequency is limited by the London penetration depth < 200 nm. It is much smaller (~ 100 times) than the separation between the generator and the detector (note an exponential decay of electromagnetic waves in the crystal with the characteristic length = skin depth, $\exp(-100) = 3.7e-44$). Furthermore, the skin depth in superconductors rapidly (quadratically, as opposed to a square root dependence for normal metals) reduces with increasing frequency. At THz frequencies, comparable with the inverse scattering time, it is negligible. This precludes propagation through the base crystal as the origin of the signal.

We added the corresponding discussion to the modified version of the manuscript.

REVIEWERS' COMMENTS:

Reviewer #1 (Remarks to the Author):

I have received and read the revised manuscript from E. A. Borodianskyi and V. M. Krasnov with the new title “Josephson emission with a record high frequency span 1-11 THz from small Bi₂Sr₂CaCu₂O₈+ mesa structures”. The authors have improved the manuscript significantly where most importantly the 3D sketch in Fig. 1 (b) gives a much better overview of the structure and the inclusion of a different emitter/detector configuration including frequencies down to 1 THz and well. At the same time the manuscript now contains more details that addresses most of the comments I had to the previous version.

I still find that the manuscript overall is of very technical sound and not providing much new physics other than pushing the stacking of small mesas closer towards the theoretical limit. This is "just" technically impressive and I find it more appropriate for a more specialized journal than Nature Communications, but I will leave this decision to the editor.

My biggest concern about this paper is still how the THz emission is calibrated and the overall benchmarking of the emitter/detector. The problem about introducing a new emitter and detector based on the same technology at the time is that any artifacts might be visible and I guess this may have been the author's motivation to compare with the BWO source in the first place. But the fact that their detector mesa “barely measures” the BWO source could be for many reasons, especially the coupling and/or alignment, and I think it is not reasonable to use the “almost-zero-power” detection of the BWO as a calibration measurement (although the proof that the structure is sensitive at 1 THz certainly helps). On the positive side I do not think that the power achievement is the most central thing in this paper – it is the emission at higher THz frequencies! The spectra shown in Fig. 3 are full of spectral features that are expected to originate from the geometrical resonances, but the peaks very different characteristics which complicates the spectra. In relation to the argument above I think this spectrum should be verified by a conventional detection technique (fx. FTIR).

I minor comment is that I do agree with the authors that the nearest competitor to the mesas are the QCLs and the pulsed sources are not so relevant. Therefore it might make more sense to include some references showing “what can today's best QCLs do” rather than the ones for pulsed THz sources at higher THz frequencies. I mostly mentioned the pulsed sources to emphasize that in principle the gap is covered but not mandatory to include, but I will leave this up to the authors.

Reviewer #2 (Remarks to the Author):

The authors have addressed all of my criticisms and suggestions in a detailed and satisfactory way. Additional data have been included strengthening the author's claims. The manuscript is very well written and contains important new physics. I recommend publication in its present form.

Reviewer #3 (Remarks to the Author):

reply (second round)

I have read through the revised manuscript and the reply from authors to the reviewers as well. Here, I will make a final report especially on the comments of reviewer #1, and the corresponding author's reply to the reviewer #1.

First of all, the authors considered the reviewer's comments seriously, and did reply back to the reviewer #1 very sincerely. More importantly, I could find no serious conflict except for the criticism of "an incremental development" in frequency, resulting in an implicit declination of acceptance of the manuscript, and suggesting it to other journals such as more application oriented journals. It is unfortunate that this sort of the phrase is commonly used for rejecting manuscript looking down from reviewer side. Moreover it does often mean that it is caused only by the strong polarized personal view. In fact, the development of THz frequency region has long been hindered as a most difficult area in a spectrum of light until just recently. In such circumstances, the intrinsic Josephson junctions are the most promising quantum system recently rapidly developing after discovery about ten years ago. The frequency range the authors obtained from 1 to 1 THz in a smaller mesa is really remarkable in this respect by any means. This observation certainly deserves as an invaluable development in this field.

I think that all replies to all reviewers, in particular, to the reviewer #1, are constructive and acceptable without having serious problems. Furthermore, the first round reviewing process brought the manuscript considerable improvement of the quality of the manuscript and, moreover it made the physical point more transparent. Therefore, I would like to judge that the revised manuscript is to be accepted for publication in Nature Communications without further evaluating each word and phrase one by one.

Second reply to Reviewers

Reply to Reviewers # 2 and 3

We want to thank the Reviewers for appreciation of our work and encouraging remarks. We all need this type of response from time to time :)

Reply to Reviewer #1

1. Reviewer writes:

“I have received and read the revised manuscript from E. A. Borodianskyi and V. M. Krasnov with the new title “Josephson emission with a record high frequency span 1-11 THz from small Bi₂Sr₂CaCu₂O₈+ mesa structures”. The authors have improved the manuscript significantly where most importantly the 3D sketch in Fig. 1 (b) gives a much better overview of the structure and the inclusion of a different emitter/detector configuration including frequencies down to 1 THz and well. At the same time the manuscript now contains more details that addresses most of the comments I had to the previous version.

I still find that the manuscript overall is of very technical sound and not providing much new physics other than pushing the stacking of small mesas closer towards the theoretical limit. This is “just” technically impressive and I find it more appropriate for a more specialized journal than Nature Communications, but I will leave this decision to the editor.”

Reply 1:

The term “technical” is not necessarily bad for a scientific work. In our eyes it means that we provide enough details for the reader to understand our message and be able to reproduce our results. Which is only good.

2. Reviewer writes:

“My biggest concern about this paper is still how the THz emission is calibrated and the overall benchmarking of the emitter/detector. The problem about introducing a new emitter and detector based on the same technology at the time is that any artifacts might be visible and I guess this may have been the author’s motivation to compare with the BWO source in the first place. But the fact that their detector mesa “barely measures” the BWO source could be for many reasons, especially the coupling and/or alignment, and I think it is not reasonable to use the “almost-zero-power” detection of the BWO as a calibration measurement (although the proof that the structure is sensitive at 1 THz certainly helps). On the positive side I do not think that the power achievement is the most central thing in this paper – it is the emission at higher THz frequencies! The spectra shown in Fig. 3 are full of spectral features that are expected to originate from the geometrical resonances, but the peaks very different characteristics which complicates the spectra. In relation to the argument above I think this spectrum should be verified by a conventional detection technique (fx. FTIR).”

Reply 2:

We basically agree with the Reviewer, as we already said in the previous round. Yet, the presence of emission is clear. The overall power is less certain, but we are not claiming any unrealistic numbers by saying that the order of magnitude is 1 μ W. In fact, we believe that this is a conservative estimation (the statement from the 1st version of the manuscript, removed later to soften the claim) because the performed calibration with the BWO provides only the lower limit estimation. More importantly, as the referee writes, the main advance of this work is not in the record power, but in the record frequency range.

3. Reviewer writes:

“I minor comment is that I do agree with the authors that the nearest competitor to the mesas are the QCLs and the pulsed sources are not so relevant. Therefore it might make more sense to include some references showing “what can today’s best QCLs do” rather than the ones for pulsed THz sources at higher THz frequencies. I mostly mentioned the pulsed sources to emphasize that in principle the gap is covered but not mandatory to include, but I will leave this up to the authors.”

Reply 3:

We agree with the referee. In the modified version we included extra two references on QCLs and an additional discussion of the differences with QCLs in the end of the first paragraph of the Discussion section:

“The reported here ten-fold frequency tunability 1-11 THz is by far more superior to that achieved in semiconductor quantum cascade lasers [5-8]. In contrast to the frequency-comb method of tuning of such lasers [7], in which the overall emitted power is spread over the entire frequency comb, in our case practically all the emission power is concentrated in the (single) primary Josephson frequency with only a minor fraction going to higher harmonics [20, 27]. Therefore, superconducting THz sources are superior in terms of tunability because the extreme broad range tunability of the primary frequency occurs without loss of radiation efficacy.”